# The cytoplasmic nuclear receptor RARγ controls RIP1 initiated cell death when cIAP activity is inhibited

Qing Xu[1], Siriporn Jitkaew[1,2], Swati Choksi[1], Chamila Kadigamuwa[1], Jianhui Qu[1], Moran Choe[1], Jonathan Jang[1], Chengyu Liu[3] & Zheng-gang Liu[1]

Tumor necrosis factor (TNF) has a critical role in diverse cellular events including inflammation, apoptosis and necroptosis through different signaling complexes. However, little is known about how the transition from inflammatory signaling to the engagement of death pathways is modulated. Here we report that the cytoplasmic retinoic acid receptor gamma (RARγ) controls receptor-interacting protein kinase 1 (RIP1)-initiated cell death when cellular inhibitor of apoptosis (cIAP) activity is blocked. Through screening a short hairpin RNA library, we found that RARγ was essential for TNF-induced RIP1-initiated apoptosis and necroptosis. Our data suggests that RARγ initiates the formation of death signaling complexes by mediating RIP1 dissociation from TNF receptor 1. We demonstrate that RARγ is released from the nucleus to orchestrate the formation of the cytosolic death complexes. In addition, we demonstrate that RARγ has a similar role in TNF-induced necroptosis in vivo. Thus, our study suggests that nuclear receptor RARγ provides a key checkpoint for the transition from life to death.

---

[1] Center for Cancer Research, National Cancer Institute, 37 Convent Drive, Bethesda, MD 20892, USA. [2] Faculty of Allied Health Sciences, Department of Clinical Chemistry, Chulalongkorn University, Bangkok 10330, Thailand. [3] National Heart Lung and Blood Institute, National Institutes of Health, 37 Convent Drive, Bethesda, MD 20892, USA. Qing Xu and Siriporn Jitkaew contributed equally to this work. Correspondence and requests for materials should be addressed to Z.-g.L. (email: zgliu@helix.nih.gov)

The inflammatory cytokine tumor necrosis factor (TNF) induces diverse cellular responses including apoptosis and necroptosis[1–3]. The molecular mechanism of TNF signaling has been intensively investigated. It is known that TNF triggers the formation of a TNF receptor 1 (TNFR1) signaling complex by recruiting several effectors such as TNFR1-associated death domain protein (TRADD), receptor-interacting protein kinase 1 (RIP1) and TNFR-associated factor 2 (TRAF2) to mediate the activation of the transcription factor nuclear factor-κB (NF-κB) and mitogen-activaed protein (MAP) kinases[1, 3]. Importantly, under certain conditions, this TNFR1 signaling complex (complex I) dissociates from the receptor and recruits other proteins to form different secondary complexes for apoptosis and necroptosis[4–6]. It is known now that necroptosis needs RIP3 and mixed lineage kinase-domain-like (MLKL) in the necrosome[7–12]. Apoptosis is initiated through the recruitment of the death domain protein Fas-associated death domain protein (FADD) to form complex II. FADD then recruits the initiator cysteine protease Caspases-8[1, 13]. The physiological roles of these death proteins and the cross-talk between necroptosis and apoptosis have been elegantly demonstrated recently in animal models[14–20].

Both TRADD and RIP1 proteins have a death domain and interact with TNFR1 directly[21]. TNF can induce cell death through either TRADD- or RIP1-initiated pathways[22, 23]. It has been shown that TNF triggers TRADD-mediated apoptosis when de novo protein synthesis is inhibited, but engages RIP1-initiated apoptosis when RIP1 ubiquitination by E3 ligases baculoviral inhibitor of apoptosis (IAP) repeat-containing protein (IAP1/2) is blocked[22]. However, both TRADD- and RIP1-initiated cell death becomes necroptotic when caspase activity is suppressed[8, 24]. In the case of de novo protein synthesis inhibition, TRADD needs to recruit RIP1 to mediate TNF-induced necroptosis[6]. RIP1-initiated cell death also occurs in cells in response to other death factors such as Fas ligand (FasL) and TNF-related apoptosis-inducing ligand (TRAIL)[25–27]. Although some proteins such as cylindromatosis (CYLD) and cellular FLICE-like inhibitory protein (cFLIP) have been suggested to havea role in regulating the formation of complex II/necrosome[1, 28], little is known about how the transition from the TNFR1 complex to the cell death complexes is modulated.

Retinoic acid receptors (RARs), RARα, RARβ and RARγ belong to the super family of nuclear hormone receptor and act as transcription factors after activation by RA[29, 30]. RARs regulate the expression of a large number of genes that are critical for cell growth, differentiation and cell death[31]. Although the localization of these RARs is predominantly nuclear, however, cytoplasmic localizations of RARs have been reported in some types of cells, but the function of the cytosolic RARs is unknown[32].

Here we report that RARγ has a critical role in RIP1-, but not TRADD-, initiated cell death in response to TNF and other death factors treatment. We found that RARγ is released from the nucleus to orchestrate the formation of the cytosolic cell death complexes. Our findings suggest that the nuclear receptor RARγ functions as a critical checkpoint of RIP1-initiated cell death.

## Results

**RARγ is required for cell death initiated by RIP1.** To identify additional components of TNF-induced necroptosis, we used a retroviral short hairpin RNA (shRNA)-mediated genetic screen to identify genes whose knockdown resulting in resistance to necroptosis. The pseudo-kinase protein MLKL was identified as a key mediator of necroptosis through screening a kinase/phosphatase shRNA library[11]. Another shRNA library used in our screening is one targeting cancer-implicated genes and this library

of 1,841 shRNAs targets 1272 human genes[33]. HT-29 cells were infected with the retroviral shRNA library and were treated to undergo necroptosis by the combination of TNF-α, Smac mimetic and the caspase inhibitor z-VAD-fmk (TSZ) (Supplementary Fig. 1). Surviving cell clones were selected for confirmation of necrotic resistance and for identification of the corresponding shRNAs by PCR and DNA sequencing. Among the 60 selected clones, 7 clones had the shRNA targeting the rarγ gene.

RARγ has two major splicing isoforms and the full length RARγ is a 454-amino acid protein[34]. The seven isolated RARγ-shRNA clones had the same shRNA that targets the 3′-untranslated region 2,680–2,700 of rarγ. Three of the seven clones (RARγ-shRNA-A, -B and -C) were used for further testing. Compared with the control firefly shRNA cells (cont-shRNA), all three RARγ-shRNA clones were resistant to necroptosis (Fig. 1a). The necroptosis resistance of these three RARγ-shRNA clones is comparable to that of RIP1-shRNA cells. The expression levels of RARγ, RARα, RIP1, RIP3, TRADD, MLKL, CYLD and TRAF2 in the three RARγ-shRNA clones and cont-shRNA cells were examined by western blotting (Fig. 1a) and the results indicated that the RARγ-shRNA specifically knocked down the level of RARγ protein and did not affect the expression of other related proteins. RARγ-shRNA-A clone was then used as a representative in the rest of the experiments. To rule out off-target effect of the RARγ-shRNA on necroptosis, a RARγ-shRNA resistant RARγ protein (rRARγ) was introduced back into the RARγ-shRNA-A HT-29 cells. The ectopic expression of the resistant RARγ restored the sensitivity of the cells to TNF-induced necroptosis, indicating that the resistance of RARγ-shRNA-A cells to necroptosis is specifically due to the knockdown of RARγ (Fig. 1b). The specific contribution of RARγ in necroptosis was further confirmed by using two different RARγ-shRNAs (RARγ-shRNA-1 and RARγ-shRNA-2) in a variety of cell lines (Supplementary Fig. 2). Importantly, knockdown of RARγ had no effect on TNF-induced activation of NF-κB and MAP kinase, Janus kinase and TSZ-induced RIP1 autophosphorylation (Supplementary Fig. 3).

To examine whether RARγ is specifically required for necroptosis, we treated these cells with TNF-α and Smac mimetic (TS) to induce RIP1-initiated apoptosis. As previously shown[22], RIP1-shRNA cells were resistant to TS-induced apoptosis. Surprisingly, RARγ-shRNA cells behaved similarly and were resistant to TS-induced apoptosis compared with cont-shRNA cells (Fig. 1c). We then examined whether RARγ is involved in TNF-induced TRADD-initiated apoptosis and necroptosis by treating these cells with TNF-α plus the protein synthesis inhibitor cycloheximide with/out zVAD (TC or TCZ). Knockdown of RARγ had little effect on TC-induced apoptosis or TCZ-induced necroptosis while RIP1 knockdown sensitized cells to TC-induced apoptosis, but rendered the cells more resistant to TCZ-induced necroptosis (Fig. 1d). Therefore, these results suggest that RARγ is likely involved in TNF-induced RIP1-initiated apoptosis and necroptosis. Similar observations were made in several other cell lines (Supplementary Fig. 2). To further verify this, we then generated TRADD knockdown cells in cont-shRNA and RARγ-shRNA HT-29 cells (Fig. 1e). The cells with TRADD knockdown were still sensitive to cell death by TS or TSZ although their sensitivity was slightly decreased (Fig. 1f). Importantly, RARγ knockdown protects cells from cell death in RARγ-shRNA-A + TRADD-shRNA cells. Therefore, RARγ is specifically required for TNF-induced RIP1-initiated apoptosis and necroptosis.

In addition, although RARα is previously reported to be involved in apoptosis[35], we found that RARα knockdown had no effect on the sensitivity of HT-29 cells to TSZ-induced RIP1-initiated apoptosis or necroptosis (Supplementary Fig. 4a).

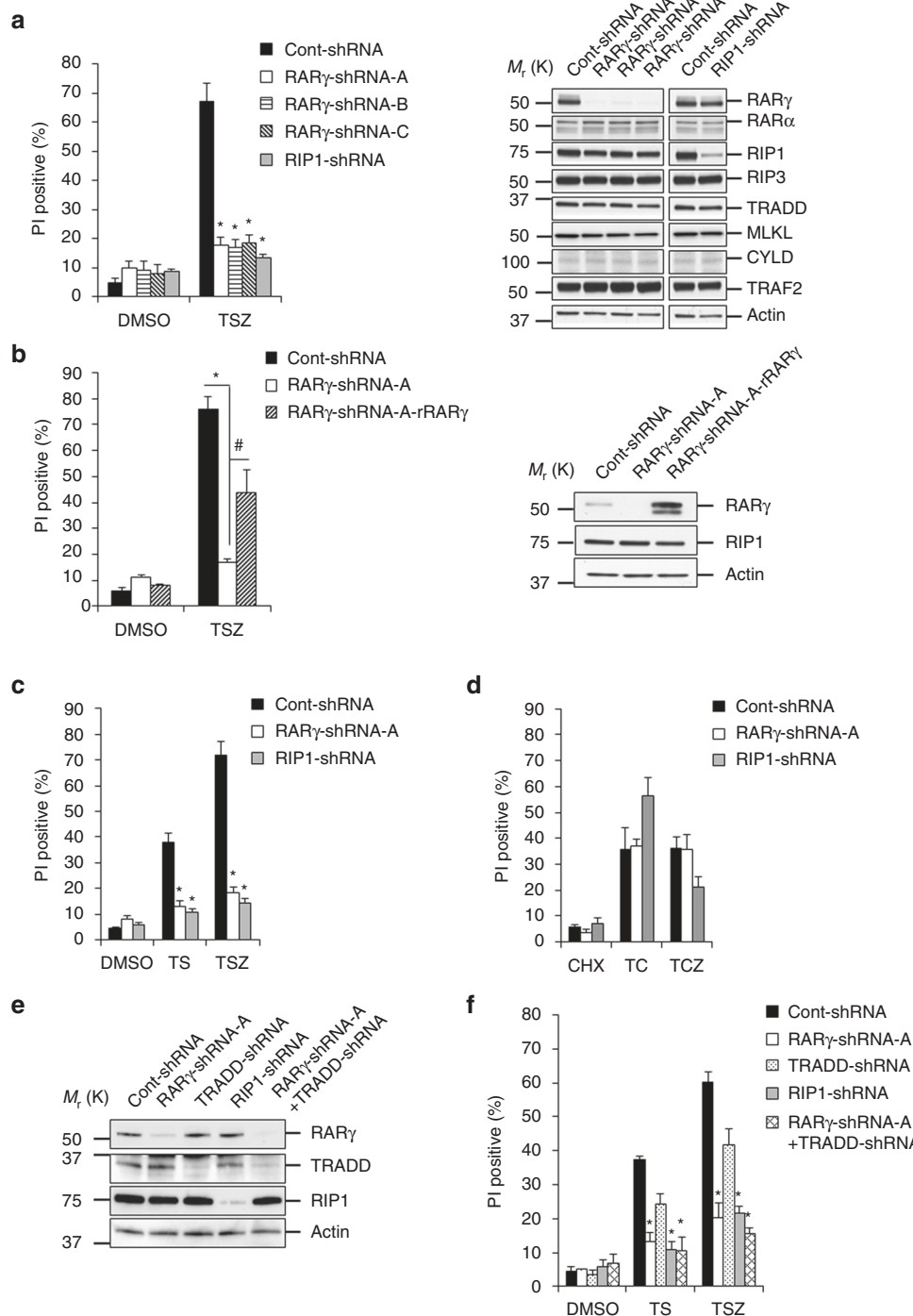

**Fig. 1** RARγ is required for TNF-induced RIP1-dependent apoptosis and necroptosis. **a** Cell death analysis of HT-29 clones cont-shRNA, RARγ-shRNA-A, RARγ-shRNA-B, RARγ-shRNA-C or RIP1-shRNA when treated with TSZ for 24 h was determined by PI staining using flow cytometry (*left panel*) (*$P < 0.05$ versus cont-shRNA; ANOVA). The bars represent the mean ± s.e.m. of three experiments. Western blot analysis of cells as mentioned in *left panel*; cell lysates were probed with antibodies as indicated (*right panel*). **b** Cell death determination by PI staining using flow cytometry of HT-29 cont-shRNA, RARγ-shRNA-A or RARγ-shRNA-A cells reconstituted with RARγ-shRNA resistant RARγ protein (RARγ-shRNA-A-rRARγ) when treated with DMSO or TSZ for 24 h (*left panel*) (*$P < 0.05$ versus cont-shRNA. #$P < 0.05$ versus RARγ-shRNA-A; ANOVA). The bars represent the mean ± s.e.m. of three experiments. Western blot analysis of cells as mentioned in left panel; cell lysates were probed with antibodies as indicated (*right panel*). **c, d** HT-29, RARγ-shRNA-A, or RIP1-shRNA cells were treated with necrotic (DMSO, TS, or TSZ) **c** or apoptotic (CHX, TC or TCZ) **d** conditions for 24 h. PI positive population was determined by flow cytometry. (*$P < 0.05$ versus cont-shRNA; ANOVA). The bars represent the mean ± s.e.m. of three experiments. **e** Western blot analysis of HT-29 cont-shRNA cells infected with TRADD-shRNA, or RARγ-shRNA-A cells infected with TRADD-shRNA lentivirus (RARγ-shRNA-A + TRADD-shRNA). Cell lysates were probed with antibodies as indicated. **f** Cells from **e** were treated with DMSO, TS, or TSZ for 24 h and the PI-positive population was determined by flow cytometry. (*$P < 0.05$ versus cont-shRNA; ANOVA). The *bars* represent the mean ± s.e.m. of three experiments. All blots above are representative of one of three experiments

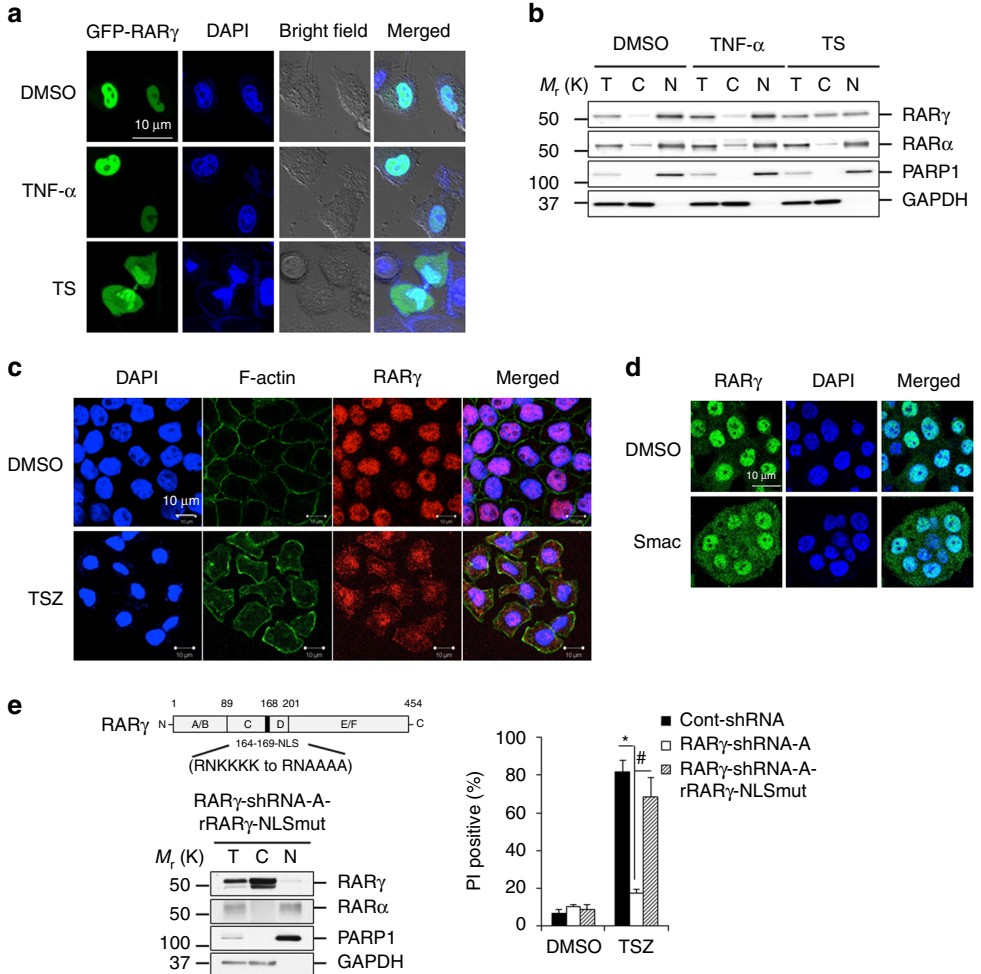

**Fig. 2** Cytosolic RARγ mediates apoptosis and necroptosis. **a** Confocal microscopy of HeLa cells transfected with RARγ-GFP plasmid and treated with DMSO, TNF-α or TS for 2 h. (*blue* DAPI; *green*: RARγ) (*scale bar*: 10 μm). **b** The total (T), cytosolic (C) and nuclear (N) fractions analysis of HeLa cells treated with DMSO, TNF-α or TS for 4 h. Fractions were analyzed by immunoblotting with the indicated antibodies. **c** Confocal microscopy of HT-29 cells treated with DMSO or TSZ for 2 h. Cells were stained with nuclear DAPI (*blue*), anti-F-actin (*green*) and anti-RARγ (*red*) antibodies (*scale bar*: 10 μm). **d** Confocal microscopy of HT-29 cells treated with Smac-minic for 2 h. Cells were stained with nuclear DAPI (*blue*) and anti-RARγ (*green*) antibodies (*scale bar*: 10 μm). **e** Scheme for the mutation of RARγ in NLS (*upper left panel*). The total (T), cytosolic (C) and nuclear (N) fractions of HT-29 RARγ-shRNA-A cells infected with resistant RARγ-NLS mutation lentiviral plasmid (RARγ-shRNA-A-RARγ-NLSmut). The cell lysates were by immunoblotted with the indicated antibodies (*lower left panel*). HT-29 cont-shRNA, RARγ-shRNA-A or RARγ-shRNA-A-rRARγ-NLSmut cells were treated with TSZ for 24 h and PI-positive population was determined by flow cytometry (*right panel*). (\*$P < 0.05$ versus cont-shRNA; #$P < 0.05$ versus RARγ-shRNA-A; ANOVA). The *bars* represent the mean ± s.e.m. of three experiments. All images and blots above are representative of one of three experiments

Also, RARα does not have any redundant role with RARγ in the process since knockdown of RARα in RARγ-shRNA-A cells did not further increase the resistance of the cells to TNF-induced RIP1-initiated apoptosis and necroptosis (Supplementary Fig. 4b).

**Cytosolic RARγ mediates cell death initiated by RIP1.** RARγ localizes in the nucleus as a ligand-dependent transcription factor, but may have a cytosolic localization[32]. We next examined the cellular localization of RARγ during the process of cell death. We first transfected HeLa cells with a green fluorescent protein (GFP)-tagged RARγ plasmid and found that the GFP-RARγ protein almost exclusively localized in the nucleus (Fig. 2a and Supplementary Fig. 5). Although TNF-α alone did not have any effect on the localization of RARγ, TS treatment caused a dramatic increase of GFP-RARγ presence in the cytoplasm (Fig. 2a and Supplementary Figs 5 and 6). Importantly, GFP-RARα remained predominantly in the nucleus under TS treatment (Supplementary Fig. 7). The increased cytosolic localization of RARγ during RIP1-initiated apoptosis was further verified by examining the endogenous RARγ protein by nuclear and cytosolic fractionation experiments. The cytosolic level of the endogenous RARγ protein was significantly higher, while the nuclear RARγ amount was reduced, in the TS-treated HeLa cells compared with the non-treated control or TNF-treated cells (Fig. 2b). Nuclear protein PARP1 and cytosolic protein GAPDH were used as controls in this experiment. The cytosolic localization of RARγ was also observed during RIP1-initiated necroptosis by immunostaining the endogenous RARγ in TSZ-treated HT-29 cells. Although the endogenous RARγ protein mainly localized in the nucleus in non-treated cells, the level of cytosolic RARγ was dramatically increased in TSZ-treated cells (Fig. 2c). Also, inhibition of TSZ-induced necroptosis by the specific RIP1 inhibitor, Necrostatin-1, had no effect on the dramatic increase of RARγ localization in the cytoplasm (Supplementary Fig. 8). These results indicated that RARγ was released to the cytoplasm from the nucleus during RIP1-initiated apoptosis and necroptosis.

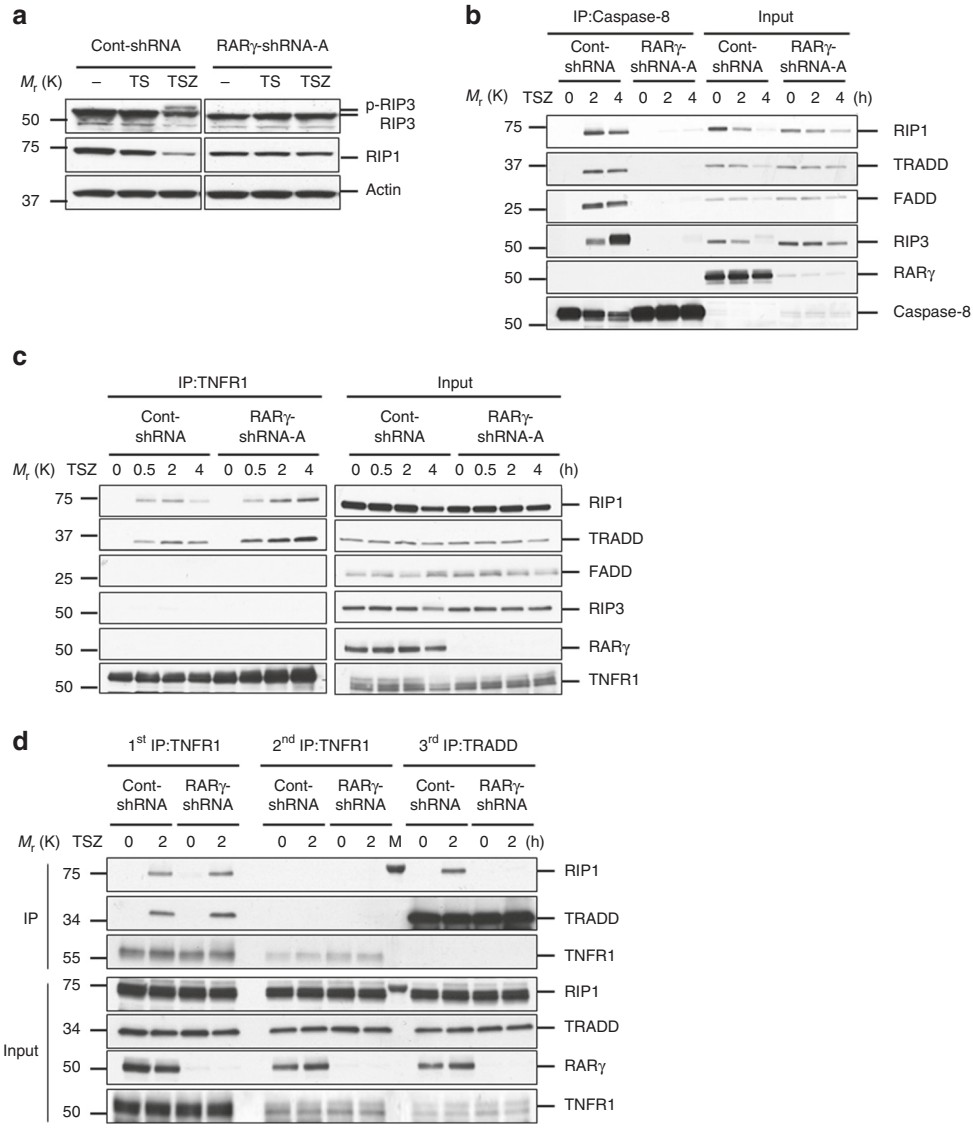

**Fig. 3** RARγ has a role in the formation of death complexes. **a** Western blot analysis of HT-29 cont-shRNA, and RARγ-shRNA-A treated with DMSO, TS or TSZ for 24 h and cell lysates were immunoblotted with the indicated antibodies. **b**, **c** Immunoprecipitation of HT-29 cont-shRNA or RARγ-shRNA-A cells treated with TSZ for the indicated time. Cell lysates were collected and immunoprecipitated with anti-Caspase-8 antibody **b** or anti-TNFR1 antibody **c**. The immunoprecipitated complexes were immunoblotted with the indicated antibodies. **d** Sequential immunoprecipitation of HT-29 cont-shRNA or RARγ-shRNA-A cells treated TSZ for 2 h. *First IP*: TNFR1 complex I was immunoprecipitated using anti-TNFR1 antibody. *Second IP*: the remaining lysates were immunoprecipitated again with anti-TNFR1 antibody. *Third IP*: the remaining lysates were then immunoprecipitated with anti-TRADD antibody. The immunoprecipitated complexes were analyzed by with the indicated antibodies. (M: marker). All blots above are representative of one of three experiments

Interestingly, Smac-mimetic treatment alone was sufficient to trigger the release of RARγ from the nucleus (Fig. 2d and Supplementary Fig. 9a, b). As TNF treatment alone without Smac mimetic induces necroptosis in mouse L929 cells[36] (Supplementary Fig. 2c), we then examined RARγ localization in L929 cells and found that unlike in HT-29 and HeLa cells, RARγ, but not RARα, localizes in both the cytoplasm and the nucleus in the non-treated cells (Supplementary Fig. 10). Also, it is known that TZ treatment without Smac mimetic induces necroptosis in bone marrow-derived macrophages (BMDMs)[9]. We then examined RARα and RARγ localizations in BMDMs and found that RARγ localizes in both the cytoplasm and the nucleus, but RARα localizes only in the nucleus (Supplementary Fig. 11a). We also found that the protein levels of cIAP2 are low in BMDMs in comparison with mouse embryonic fibroblasts (MEFs)

(Supplementary Fig. 11b). We next examined whether cellular IAPs (cIAPs) are required for RARγ nuclear localization. Although loss of cIAP1 has limited effect on RARγ localization, the dominant negative cIAP2 mutant, cIAP2[H570A], which blocks both cIAP1 and cIAP2 function[37], enables RARγ to localize in both the cytoplasm and the nucleus, indicating that cIAPs may be required for RARγ nuclear localization (Supplementary Fig. 12a). Consistently, TZ treatment without Smac mimetic triggers necroptosis in the cIAP2[H570A] MEFs (Supplementary Fig. 12b). Taken together, these results suggest that the release of RARγ from the nucleus by Smac mimetic is most likely through inhibiting cIAPs function.

To address whether the cytosolic or the nuclear RARγ has a role in TNF-induced cell death, we mutated the RARγ nuclear localization signal (NLS) in the RARγ-shRNA-resistant RARγ

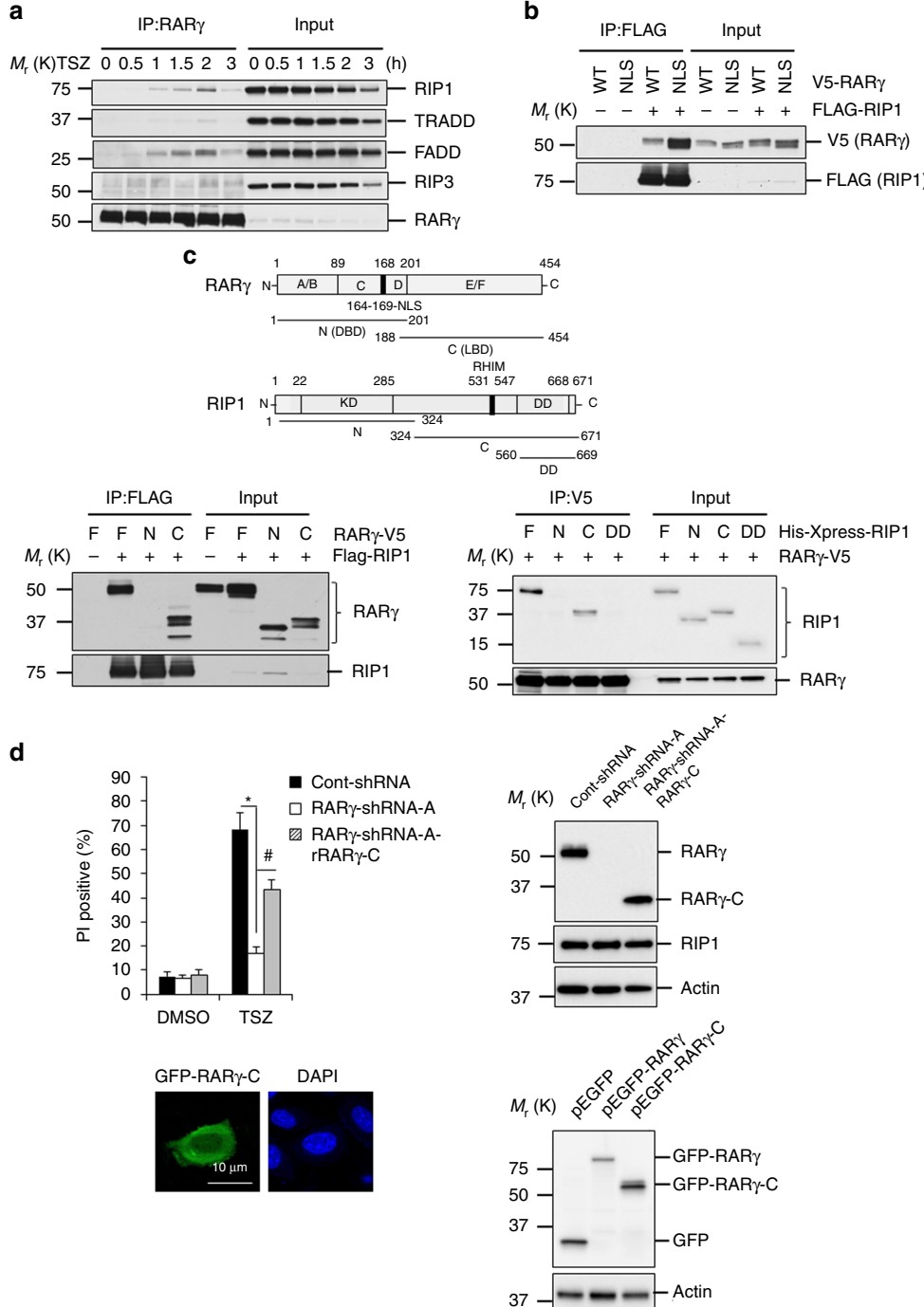

**Fig. 4** RARγ C fragment are important for protecting cell from TNF-induced necroptosis via interaction with RIP1. **a** Immunoprecipitation of HT-29 cells treated with TSZ for the indicated times. Cell lysates were immunoprecipitated with anti-RARγ antibody and analyzed by immunoblotting with the indicated antibodies. **b** HEK293T cells were co-transfected with wild-type V5-RARγ (WT) or the V5-RARγ-NLSmut (NLS) and with or without FLAG-RIP1 plasmids. Cell lysates were immunoprecipitated using anti-FLAG (RIP1) antibody and analyzed with the indicated antibodies. **c** Scheme of RARγ and RIP1 genes (*upper panel*). HEK293T cells were co-transfected with different fragment of RARγ-V5 (F: full; N:1–201aa; C: 188–454aa) and with or without FLAG-RIP1 plasmids; or with different fragment of His-Xpress-RIP1 (F: full; N: 1–324aa; C: 324–671aa; DD: 560–669aa) and with or without RARγ-V5. Cell lysates were immunoprecipitated using anti-FLAG (RIP1) or anti-V5 (RARγ) antibody and analyzed with the indicated antibodies. **d** HT-29 RARγ-shRNA-A cells infected with rRARγ-C. Cell death analysis of HT-29 cont-shRNA, RARγ-shRNA-A, and RARγ-shRNA-A + rRARγ-C when treated with TSZ for 24 h was determined by PI staining using flow cytometry (*upper left panel*). (*$P < 0.05$ versus cont-shRNA; #$P < 0.05$ versus RARγ-shRNA-A; ANOVA). The *bars* represent the mean ± s.e.m. of three experiments. Western blot analysis of cells as mentioned in left panel (*upper right panel*). Confocal microscopy of HT-29 cells infected with GFP-RARγ-C plasmid (*lower left panel*) (*blue*: DAPI; *green*: RARγ). Western blot analysis of HEK293T cells were transfected with p-EGFP, p-EGFP-RARγ and p-EGFP-RARγ-C by using anti-GFP and anti-Actin antibodies (*lower right panel*). All blots and images above are representative of one of three experiments

plasmid (rRARγ-NLSmut). When this rRARγ-NLSmut protein was introduced back into the RARγ-shRNA-A HT-29 cells, the mutant RARγ localized in the cytoplasm and was able to fully rescue TSZ-induced necroptosis (Fig. 2e). These data suggested that the cytosolic RARγ is critical for TSZ-induced necroptosis.

**RARγ is essential for formation of cytosolic death complexes.** RIP3 autophosphorylation is an early event in TNF-induced necroptosis[7–9]. In RARγ knockdown cells, RIP3 phosphorylation was abolished (Fig. 3a). This result indicated that RARγ functions upstream of RIP3 phosphorylation during necroptosis. We then examined whether RARγ is required for the formation of the

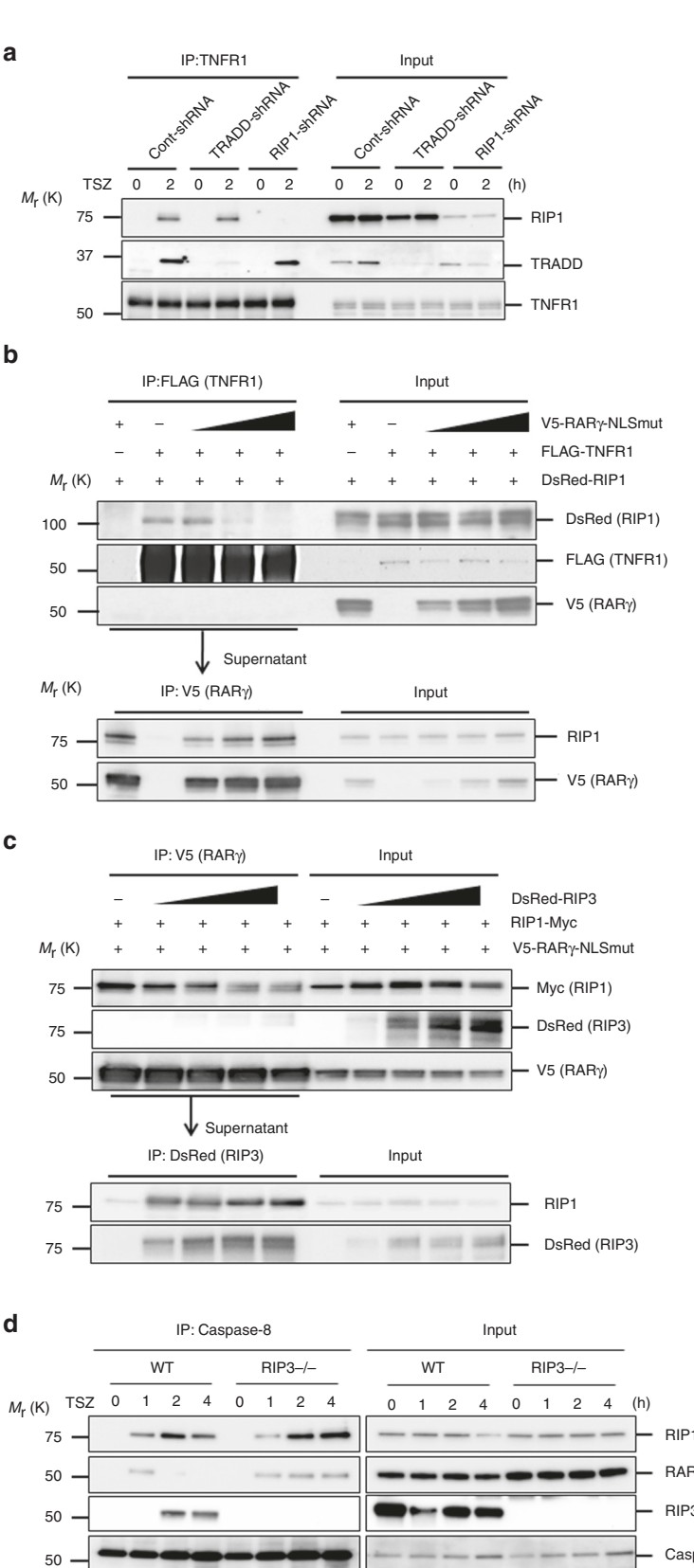

necrosome. It is known that the necrosome consists of multiple proteins including RIP1, RIP3, FADD and Caspase-8, and can be detected by immunoprecipitating Caspase-8[7–9]. HT-29 cells expressing RARγ-shRNA or cont-shRNA were treated with TSZ and collected for immunoprecipitation with a Caspase-8 specific antibody. The necrosome components, RIP1, RIP3 and FADD were efficiently pulled down by immunoprecipitating Caspase-8 in TSZ-treated cont-shRNA cells, but the amounts of RIP1, RIP3 and FADD proteins pulled down by Caspase-8 immuno-precipitation were almost unnoticeable in RARγ-shRNA cells (Fig. 3b). Consistent with previous reports[6], some TRADD protein was also detected in the necroptotic complex in the control cells. However, no RARγ protein was found in the same complex. These results suggested that RARγ knockdown prevented the formation of the necrosome. We found that RARγ is also required for the formation of apoptotic complex IIa (Supplementary Fig. 13).

It is believed that TRADD and RIP1 dissociate from the receptor and recruit other proteins to form the secondary death complexes[1, 8, 38]. As RARγ is required for the formation of the necrosome, but is not a part of the complex, we next examined whether RARγ has any role in the formation of the TNFR1 complex by immunoprecipitating TNFR1. RARγ knockdown has no effect on the formation of TNFR1 complex, because TNFR1 immunoprecipitation efficiently pulled down RIP1 and TRADD proteins in both con-shRNA and RARγ-shRNA cells (Fig. 3c). Consistent with previous findings[6, 8], no FADD and RIP3 proteins were found in the TNFR1 complex. Again, no RARγ protein was detected in the TNFR1 complex. These results indicated that RARγ is not required for the assembling of the TNFR1 complex. As it has been suggested that TNFR1 endocytosis may be required for downstream signaling[39], we checked whether RARγ knockdown affects TNFR1 internalization[40] and found that RARγ knockdown has no effect on TNFR1 internalization (Supplementary Fig. 14). We then investigated whether RARγ knockdown affects the dissociation of TRADD and RIP1 proteins from TNFR1 by examining the presence of the non-receptor-bound TRADD/RIP1 complex. After two sequential immunoprecipitation of TNFR1, all TNFR1-bound TRADD and RIP1 were removed from the cell lysates. Next, the TRADD immunoprecipitation with the remaining lysates pulled down RIP1 only from cont-shRNA cells, but not from RARγ-shRNA cells, indicating that the TRADD and RIP1 complex did not dissociate from TNFR1 in the absence of RARγ (Fig. 3d). Therefore, it is possible that RARγ may regulate the dissociation of complex I from the TNFR1 receptor.

As RARγ is not in either the TNFR1 complex or the necrosome, we then examined what TNF effector proteins it may interact with by immune-precipitating RARγ protein. Interestingly, both RIP1 and FADD were co-precipitated with RARγ as early as 1 h after TSZ treatment (Fig. 4a). The amount of RIP1 and FADD peaked at 2 h and decreased by 3 h after treatment. Only a marginal amount of TRADD and no RIP3 were

detected in the RARγ precipitants. These results suggested that RARγ forms a complex with RIP1 and FADD that is different from the necrosome. This observation raised the possibility that RARγ may interact with RIP1 and FADD. We then evaluated the interaction between RARγ with each of these proteins by cotransfection/immunoprecipitation experiments. Immunoprecipitating FLAG-RIP1 clearly pulled down RARγ and more efficiently the cytoplasmic RARγ-NLSmut, but none of the FADD, TRADD or RIP3 immunoprecipitation did so (Fig. 4b and Supplementary Fig. 15a–c). In addition, we found that RARα did not interact with RIP1 (Supplementary Fig. 15d). The interaction of RIP1 and RARγ was also confirmed by immunoprecipitating the endogenous RIP1 following TSZ treatment (Supplementary Fig. 16). These results suggested that RARγ specifically interacts with RIP1.

To better understand how RARγ interacts with RIP1, we generated several truncated RARγ and RIP1 proteins and examined which regions of these two proteins interact (Fig. 4c). We found that the C-terminal region of RARγ, which does not include the transcription activation domain, DNA binding domain and the NLS site, is fully capable to interact with RIP1 protein, most likely through RIP1's RIP homotypic interaction motif domain (Fig. 4c). The C-terminal region of RARγ still contains the activation function 2 region that does in fact interact with coactivator proteins to confer ligand inducible transcriptional activation to RARγ. Importantly, when a RARγ-shRNA resistant c-terminal region of RARγ was expressed in RARγ-shRNA-A cells, it localized in the cytoplasm and rescued the sensitivity of the cells to TSZ-induced necroptosis (Fig. 4d). These results further supported our conclusion that the cytosolic RARγ is required for RIP1-initiated cell death.

Both TRADD and RIP1 contribute to the formation of the TNFR1 complex[4]. To test whether TRADD is important for TNFR1 and RIP1 interaction in necroptosis, we next examined TNFR1 and RIP1 interaction in TRADD knockdown cells. Immunoprecipitating TNFR1 pulled down RIP1 similarly in the control and TRADD-shRNA cells, indicating that TRADD knockdown does not have any effect on RIP1 recruitment to TNFR1 in necroptosis (Fig. 5a). Therefore, TNFR1 and RIP1 interaction is independent of TRADD in necroptosis. Next, we tested whether RARγ affects TNFR1 and RIP1 interaction. HEK293T cells were co-transfected with FLAG-TNFR1, DsRed-RIP1 and increasing amounts of V5-RARγ-NLSmut plasmids. Immunoprecipitation of TNFR1 clearly co-precipitated RIP1 protein. However, in the presence of increasing amounts of RARγ, the TNFR1 and RIP1 interaction gradually diminished (Fig. 5b, *top panel*). When the cell lysates after TNFR1 pull-down were used to immunoprecipitate V5-RARγ, increasing levels of RIP1 were detected (Fig. 5b, *bottom panel*). These results indicated that RARγ disrupts the TNFR1 and RIP1 interaction by pulling away RIP1 protein from TNFR1.

As the RARγ, RIP1 and FADD complex seems to be transient as seen in Fig 4a, and the necrosome complex does not have the

---

**Fig. 5** RARγ initiates the formation of death complexes by dissociating RIP1 from TNFR1. **a** Immunoprecipitation of HT-29 cont-shRNA, TRADD-shRNA or RIP1-shRNA treated with TSZ for the indicated times. Cell lysates were immunoprecipitated using anti-TNFR1 antibody and immunoblotted with the indicated antibodies. **b** Sequential immunoprecipitation of HEK293T cells co-transfected with FLAG-TNFR1, DsRed-RIP1 and increasing amounts of V5-RARγ-NLSmut plasmids as indicated. *First IP*: cell lysates were immunoprecipitated with anti-FLAG (TNFR1) antibody. *Second IP*: the remaining lysates were then immunoprecipitated with anti-V5 (RARγ) antibody. The immunoprecipitated complexes were analyzed by with the indicated antibodies. **c** Sequential immunoprecipitation of HEK293T cells co-transfected with V5-RARγ-NLSmut, RIP1-Myc, and increasing amounts of DsRed-RIP3 plasmids as indicated. *First IP*: cell lysates were immunoprecipitated with anti-V5 (RARγ) antibody. *Second IP*: the remaining lysates were then immunoprecipitated with anti-DsRed (RIP3) antibody. The immunoprecipitated complexes were analyzed with the indicated antibodies. **d** WT and RIP3−/− MEFs treated with TSZ for the indicated times. Cell lysates were immunoprecipitated using anti-Caspase-8 antibody and immunoblotted with the indicated antibodies. All blots above are representative of one of three experiments

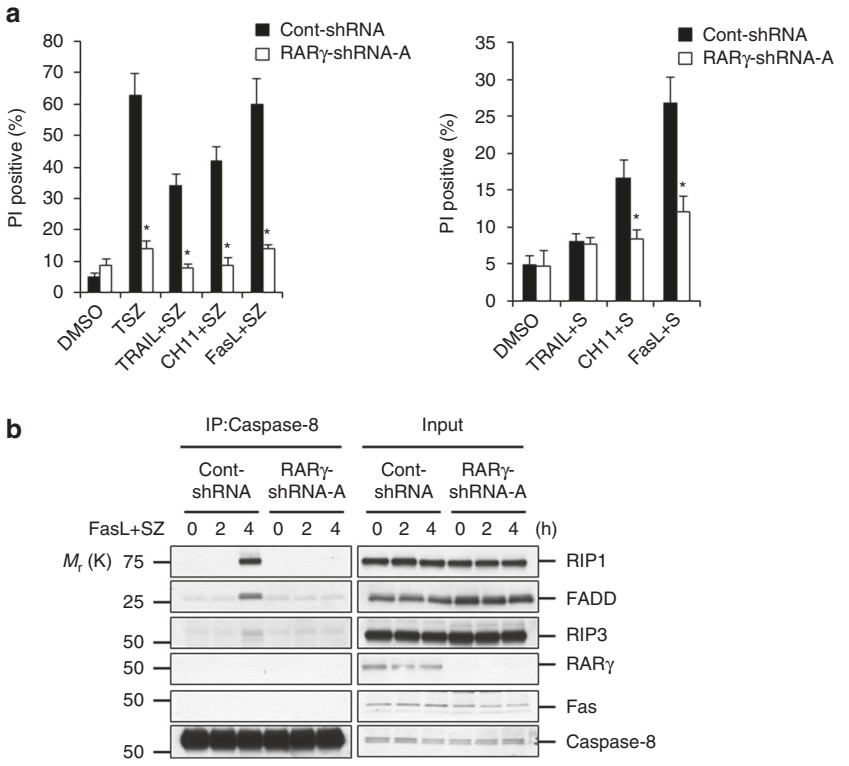

**Fig. 6** RARγ is involved in death receptor and genotoxic stresses-induced cell death. **a** Cell death analysis of HT-29 cont-shRNA and RARγ-shRNA-A treated with DMSO, TSZ, TRAIL + SZ, CH11 + SZ or FasL + SZ for 24 h (*left panel*), or treated with DMSO, TRAIL + S, CH11 + S or FasL + S for 24 h (*right panel*). Cell death was determined by PI staining using flow cytometry. (*$P < 0.05$ versus cont-shRNA; ANOVA). The bars represent the mean ± s.e.m. of three experiments. **b** HT-29 cont-shRNA, and RARγ-shRNA-A treated with FasL + SZ for the indicated times. Cell lysates were immunoprecipitated with anti-Caspase-8 antibody and analyzed with the indicated antibodies. Blots are representative of one of the three experiments

RARγ protein, we wondered whether RARγ protein was removed from the complex when other proteins such as RIP3 was recruited. We then co-transfected HEK293T cells with V5-RARγ-NLSmut, RIP1-Myc and increasing amounts of DsRed-RIP3 plasmids and immunoprecipitated V5-RARγ. The interaction between RARγ and RIP1 was significantly decreased in the presence of increasing amounts of RIP3 (Fig. 5c, *top panel*). When the cell lysates after V5-RARγ pull-down were immunoprecipitated with an anti-DsRed antibody, increasing amounts of RIP1 were pulled down (Fig. 5c, *bottom panel*). These results indicated that the RARγ and RIP1 interaction was disrupted by RIP3. To confirm that the absence of RARγ in the necrosome is due to the recruitment of RIP3, we examine whether RARγ is in the complex in RIP3−/− MEFs. In addition, as the Caspase-8 immunoprecipitation experiments (Fig. 3b) were done at 2 and 4 h time points after TSZ treatment when RIP3 is already recruited, we also checked whether RARγ could be pulled down by Caspase-8 at 1 h before RIP3 is recruited in wild-type (wt) MEFs. As shown in Fig. 5d, RARγ is indeed present in the complex at 1 h after TSZ treatment in wt cells. Importantly, while the presence of RARγ in this complex disappears at 2 and 4 h time points in wt cells, RARγ presence in the complex is sustained in RIP3−/− MEFs (Fig. 5d). These results suggest that RARγ may be a component of the TNFR1 complex IIa and is competed off when RIP3 is recruited to form necrosome/complex IIb. The presence of RARγ in the TNFR1 complex IIa was confirmed when Caspase-8 was immunoprecipitated in TSZ-treated HeLa cells (Supplementary Fig. 17), which does not have RIP3 expression[8]. Taken together, the above data suggests that RARγ initiates the formation of the necrosome by recruiting RIP1 away from TNFR1 to form the TNFR1 complex IIa.

RIP1 is known to have a role in cell death induced by death factors FasL and TRAIL[25, 26]. We examined whether RARγ is required for RIP1-initiated cell death triggered by FasL or TRAIL. Knockdown of RARγ protected cells from FasL- or TRAIL-induced cell death in the presence of the cIAP inhibitor Smac-mimemic, with/out z-VAD-fmk (Fig. 6a). RARγ knockdown also disrupted the formation of FasL-induced necrosome (Fig. 6b).

**RARγ is required for cell death initiated by RIP1 in vivo.** RARγ has two major splicing isoforms, RARγ1 and RARγ2, and RARγ1 is the major functional isoform and early studies reported that RARγ complete knockout mice have growth defect and may die prematurely[41–43]. To study the physiological role of RARγ in cell death, we generated RARγ1 knockout (RARγ1−/−) mice using the CRISPR/Cas9 system[44]. The Cas9 mRNA was co-microinjected with two single guide RNAs (sgRNAs), which target two different regions of the first coding exon of the mouse *rarγ1* gene (Fig. 7a), into the cytoplasm of mouse zygotes. The founder mice were genotyped by PCR and DNA sequencing. We selected two different founder mice that carried frame shift mutations for line expansion and subsequent studies. The line used in the below animal experiments has a ~200 bp deletion (Fig. 7b). The RARγ1-KO mice appear normal phenotypically[42].

From the RARγ1−/− mice we generated MEFs and checked for loss of RARγ1 expression and made sure that the expression levels of RIP1, RIP3, MLKL and CYLD in RARγ1−/− MEFs are not altered (Fig. 7c). The primary RARγ1−/− MEFs were resistant to TSZ-induced RIP1-initiated necroptosis while they are still sensitive to TC- or TCZ-induced TRADD-initiated cell death

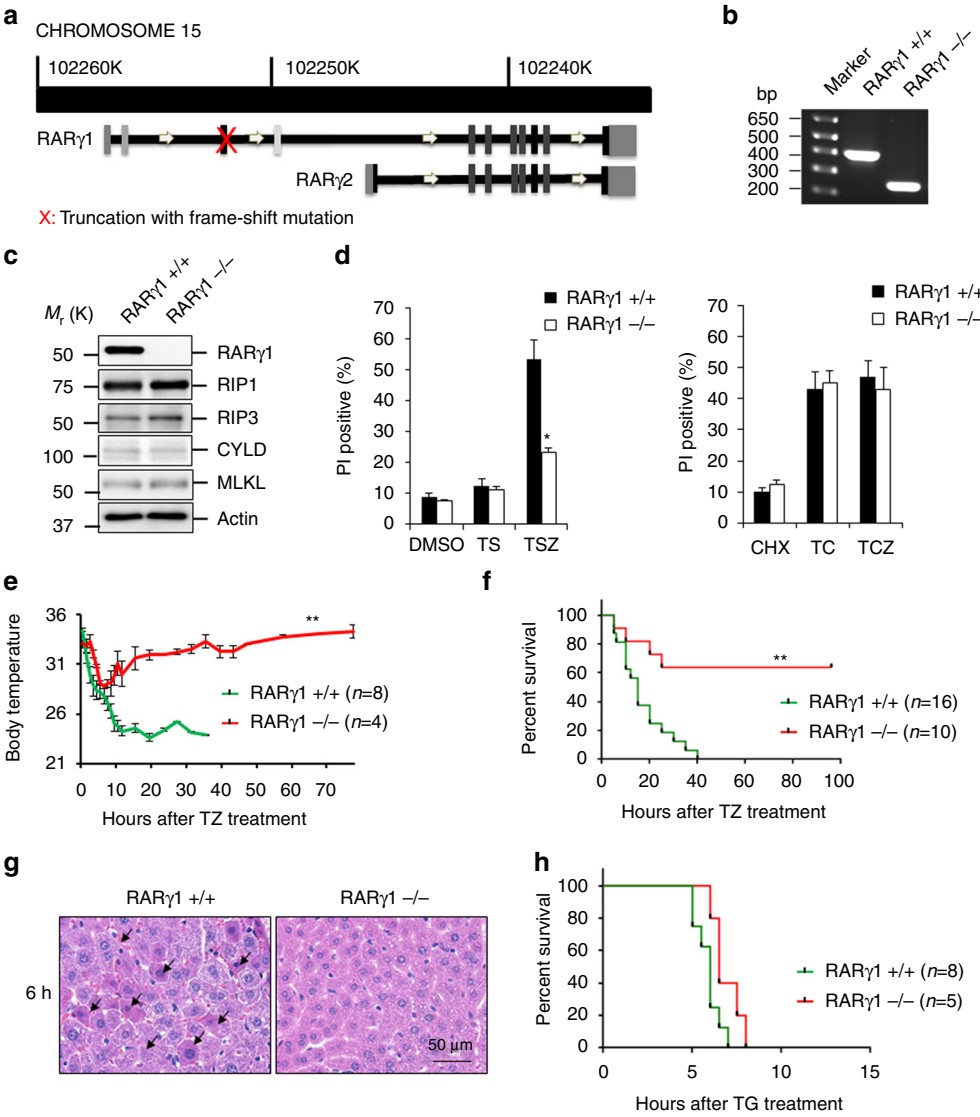

**Fig. 7** RARγ-null MEFs and mice are protected against TNF-induced death. **a** Scheme for making CRISPR-knock-out (RARγ−/−) mice. **b** PCR analysis of genomic DNA from wild-type (RARγ+/+) and homozygous CRISPR-knock-out (RARγ−/−) mice. **c** MEFs from RARγ+/+ and RARγ−/− littermates were generated and analyzed by immunoblotting using indicated antibody. **d** Primary RARγ+/+ and RARγ−/− MEFs were treated with DMSO, TS or TSZ for 24 h (*left panel*) or with CHX, TC or TCZ for 24 h (*right panel*). Cell death was determined by PI staining using flow cytometry. **e, f** Body temperature curve **e**; or survival curve **f** of wild-type (*green*: RARγ+/+) and CRISPR-knock-out (*red*: RARγ−/−) mice after treatment with z-VAD-fmk and TNF-α (TZ). (**\*\*P < 0.001** versus RARγ+/+; Student's *t*-test for **e**; log rank test for **f**. **g** Livers were excised from RARγ+/+ and RARγ−/− mice 6 h after TZ treatment. Representative histologic H&E staining of liver shows focal necrotic cells and red blood cells (*arrow*) (scale bar: 50 μm). **h** Survival curve of wild-type (*green*: RARγ+/+) and CRISPR-knockout (*red*: RARγ −/−) mice after treatment with treatment with GalN and TNF-α (no significance; log rank test). Mouse number used in each experiment was indicated. All blots and gel above are representative of one of three experiments

(Fig. 7d) and the immortalized RARγ1−/− MEFs were resistant to both TS- and TSZ-induced cell death (Supplementary Fig. 18a) confirming the results obtained with RARγ-shRNAs. In addition, RARγ1 deletion also protects the primary BMDMs cells from TSZ-induced necroptosis (Supplementary Fig. 18b). Next we used a mouse model of systemic inflammation that resembles clinical sepsis to examine the role of RARγ in cell death in vivo. It has been demonstrated that inhibition of caspases by z-VAD-fmk strongly sensitizes mice to TNF-induced shock involving RIP1/RIP3-dependent necrosis[45]. The wt and RARγ1−/− littermates were injected with a combination of TNFα and z-VAD-fmk (TZ). We monitored the body temperature of the mice for 72 h and although both wt and RARγ1−/− littermates had similar hypothermia initially, the wt mice had continued drop in body temperature whereas most RARγ1−/−

mice recovered (Fig. 7e). The wt mice succumb by 24–36 h, whereas 60% of the RARγ1−/− mice survive up to 4 days (Fig. 7f). When these mice were examined 6 and 12 h after treatment, we find that hematoxylin and eosin staining of the liver showed focal necrotic cells in wt mice, evidenced by pyknotic nuclei and eosinophilic cytoplasm whereas the RARγ1−/− livers had significantly less necrosis (Fig. 7g and Supplementary Fig. 19). Interestingly, both wt and RARγ1−/− mice have similar mortality in the TNF-induced, TRADD-mediated cell death model when mice were treated with TNF and D-galactosamine (G), which is a metabolic inhibitor of transcription (Fig. 7h)[46, 47]. In addition, TRADD−/− mice were not resistant to TZ-induced lethality (Supplementary Fig. 20). These results suggested that RARγ has a critical role in RIP1/RIP3-mediated necroptosis, but not TRADD-mediated cell death in vivo.

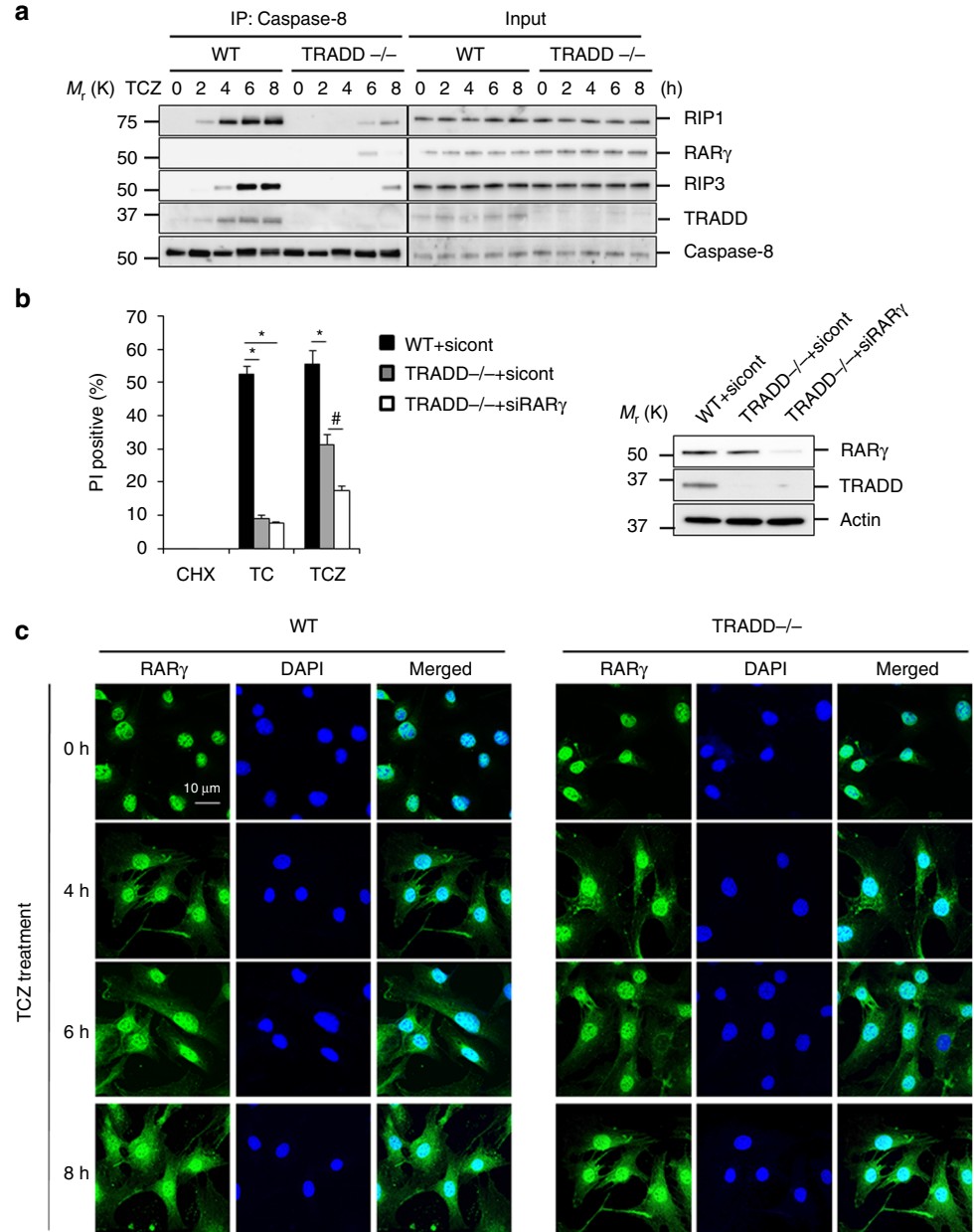

**Fig. 8** RARγ is involved in TCZ-induced necroptosis in TRADD−/− MEFs. **a** WT and TRADD−/− MEFs treated with TCZ for the indicated times. Cell lysates were immunoprecipitated using anti-Caspase-8 antibody and immunoblotted with the indicated antibodies. **b** WT or TRADD−/− MEFs transfected with sicont or siRARγ were treated with TSZ for 24 h. Cell death was determined by PI staining using flow cytometry (*left panel*). Western blot analysis of cells as mentioned in *left panel*; cell lysates were probed with antibodies as indicated (*right panel*) (*P < 0.05 versus WT + sicont. #P < 0.05 versus TRADD−/− + sicont; ANOVA). The *bars* represent the mean ± s.e.m. of three experiments. **c** WT and TRADD−/− MEFs treated with TCZ in indicated time points. The localization of endogenous RARγ assessed by immunofluorescent staining (*blue*: DAPI; *green*: RARγ) (*scale bar*: 10 μm). All blots and images above are representative of one of three experiments

**RARγ is involved in necroptosis by TCZ in TRADD null MEFs.** Our results indicated that RARγ is not required for TC- or TCZ-induced cell death, which is initiated by TRADD (Figs. 1d and 7d). However, a recent study reported that TRADD−/− MEFs are still sensitive to TCZ-induced necroptosis[48]. To investigate whether RARγ has a role in TCZ-induced necroptosis in TRADD−/− MEF cells, we first examined the formation of the TNFR1 complex IIa and necrosome in response to TCZ treatment by immunoprecipitating Caspase-8 in wt and TRADD−/− MEFs. As seen in Fig. 8a, the complex IIa components RIP1 and TRADD were co-precipitated with Caspase-8 at 2 h after treatment while the necrosome component RIP3 was detected in the complex at

4 h in wt cells. However, RIP1 and RIP3 were not detected in the Caspase-8 immunoprecipitants until 6 and 8 h after TCZ treatment in TRADD−/− MEFs (Fig. 8a). Importantly, although RARγ was not detected in either complex IIa or necrosome in wt cells, the protein was present in the complex IIa at 6 h after TCZ treatment, but not in the necrosome when RIP3 was recruited at 8 h, in TRADD−/− MEFs (Fig. 8a). These results suggest that TRADD has a major role in the formation of the TNFR1 complex IIa and necrosome in response to TCZ in wt cells and that RARγ may have a role in the process in TRADD−/− MEFs. The major role of TRADD in TCZ-induced necroptosis was supported by the fact that TRADD−/− MEFs are much less sensitive to

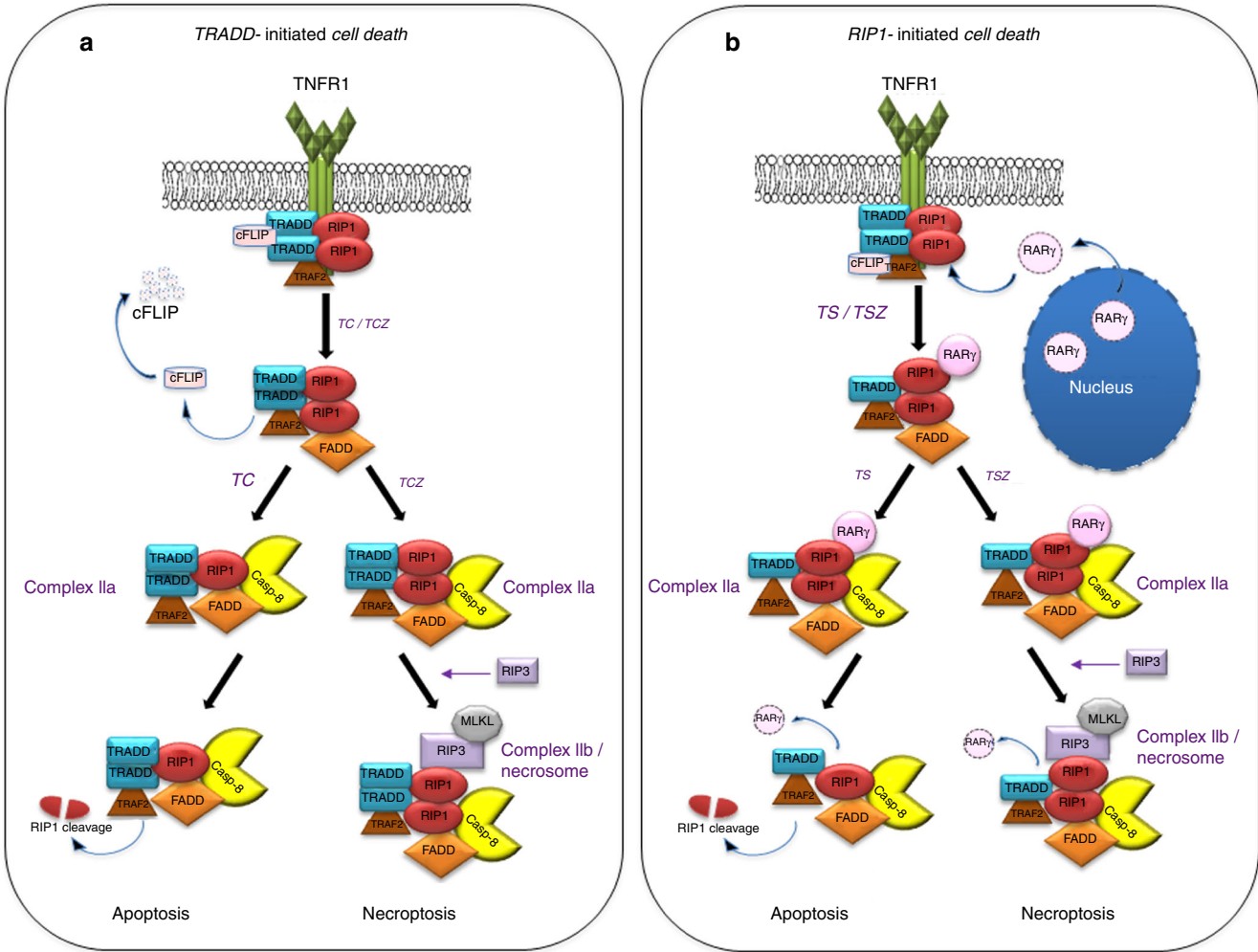

**Fig. 9** RARγ modulates TNF-induced RIP-initiated cell death. **a** TRADD is responsible for initiated TC or TCZ-induced cell death. **b** RARγ is essential for the transition from inflammatory signaling to death pathways in TNF-induced RIP1-initiated cell death. When RARγ is released from the nucleus in response to TNF under the conditions of cIAP inhibition, it initiates the formation of death complex IIa by dissociating RIP1 from TNFR1. When RIP1 recruits RIP3 to necrosome/complex IIb, RARγ is replaced from the complex

TCZ-induced necroptosis in comparison with wt cells[48] (Fig. 8b). Notably, knockdown of RARγ in TRADD−/− MEFs further reduced the sensitive of the cells to TCZ-induced necroptosis, indicating that RARγ is needed for RIP1 to mediate TCZ-induced cell death in TRADD−/− cells (Fig. 8b). Although we found that RARγ is specifically released from the nucleus by Smac mimetic (Fig. 2 and Supplementary Fig. 5–9), TCZ, but not TC, treatment resulted in its release from the nucleus nonspecifically since the treatment had the same effect on RARα (Fig. 8c and Supplementary Fig. 21).

## Discussion
Apoptosis and necroptosis are two types of programmed death that are physiologically and pathologically relevant. Although the molecular mechanism of apoptosis is well studied, the regulation of necroptosis is starting to be understood. TNF-induced apoptosis and necroptosis could be initiated by TRADD or RIP1 respectively when de novo protein synthesis is blocked or IAP E3 ligases are inhibited[1, 3]. To engage these death pathways, the formation of death complexes, complex IIa and necrosome/complex IIb, is critical, but little is known about the regulation of the formation of these complexes. RARγ was identified by selection for resistance to necroptosis. We found that the loss of RARγ rendered cells resistant to both RIP1-initiated apoptosis

and necroptosis when cIAP activity is blocked. Further, we showed that it is the cytoplasmic RARγ, not the nuclear RARγ, which mediates the formation of cytosolic death complexes. We demonstrated that RARγ interacts with RIP1 and is critical for RIP1 dissociation from the receptor. The critical role of RARγ in RIP1-initiated cell death is further supported through studying the RARγ-null cells and mice. Our data also indicated that RARγ has a broad role in cell death, because it is involved in RIP1-initiated cell death in response to other death factors, FasL and TRAIL.

To identify additional components of TNF-induced necroptosis, we used a retroviral shRNA-mediated genetic screen to identify genes whose knockdown resulting in resistance to necroptosis. In our first screening with a kinase/phosphatase shRNA library, MLKL was identified as a key mediator of necroptosis. We identified one clone of targeting RIP1 and six clones of targeting MLKL, but none of clones targeting RIP3 in this screening. Clearly, the shRNAs of each gene have different penetration/efficiency since RIP3 was not identified in our screening. We did not hit other known proteins of this pathway such as TNFR1, as they are not included in this library. In our current study, RARγ was identified by screening the shRNA library targeting cancer-related genes and this library of 1841 shRNAs targets 1,272 human genes, in which none of the

known necroptosis-related genes is included. We identified seven clones targeting RARγ in this screening.

TNF triggers the formation of TNFR1 signaling complex. Under apoptotic or necroptotic conditions, TNFR1complex dissociates from the receptor and recruits other proteins to form complex II or necrosome[1, 28]. When protein synthesis is blocked, cFLIP protein is degraded and TRADD-initiated cell death is carried out although TRADD needs to recruit RIP1 for mediating necroptosis[6] (Fig. 9a). In the case of IAP E3 ligases inhibition, RIP1-mediated cell death happens. It is thought that the blockage of RIP1 ubiquitination by cIAP1/2 is sufficient for RIP1 to recruit FADD and RIP3 to engage apoptosis and necroptosis respectively[6, 8]. However, we found that when RARγ was knocked down, the non-ubiquitinylated RIP1 was stuck in the TNFR1 complex and no complex IIa or necrosome was formed (Fig. 3 and Supplementary Fig. 13). As knocking down of RARγ does not affect TNFR1 internalization (Supplementary Fig. 14), our data suggests that RARγ is necessary for RIP1 to dissociate from the TNFR1 receptor and to initiate the formation of complex IIa and the subsequent necrosome (Fig. 4 and Supplementary Figs 13 and 17). Our in vivo study with RARγ1−/− mice further supported that RARγ is involved in RIP1-initiated necroptosis, but not in TRADD-initiated cell death, as RARγ1−/− mice are resistant to TZ, but not TG treatment (Figs 7e–h), whereas TRADD−/− mice are resistant to TG, but not TZ treatment[47] (Supplementary Fig. 20). The residual lethality of RARγ1−/− mice in response to TZ treatment may be due to the presence of RARγ2. Interesting, we found that RARγ has a role in TCZ-induced necroptosis in TRADD−/− MEFs because it is released from the nucleus non-specifically by the TCZ treatment (Fig. 8 and Supplementary Fig. 21). These findings further supported our conclusion that RARγ is required for RIP1-initiated cell death.

The functions of RARs have been extensively studied as nuclear transcription factors and the RARs are predominantly nuclear even in the absence of their ligands. In addition to their functions in regulating cell proliferation and differentiation, RARs are implicated indirectly in apoptosis through their nuclear transcriptional activity[29, 31]. However, our data indicated that the involvement of RARγ in RIP1-initiated cell death does not rely on its transcriptional activity (Figs 2 and 4c, d). Instead, RARγ has a role in RIP1-initiated cell death as a cytosolic protein. Though RARs are nuclear proteins, RARγ could be found in the cytoplasm when overexpressed and the subcellular localization of RARγ is highly regulated[32]. Our data indicated that this TS or TSZ-induced cytoplasmic localization is specific to RARγ because the localization of RARα is not altered (Fig. 2 and Supplementary Figs 5–7). Interesting, Smac mimetic alone is sufficient to induced the release of RARγ from the nucleus (Supplementary Fig. 9). Overexpression of RXRα also did not have any effect on RARγ release from the nucleus in response to TS treatment in HeLa cells (Supplementary Fig. 22). Currently, it seems that the RARγ cytoplasmic localization is induced by Smac mimetic through inhibiting cIAP1/2 function because RARγ localizes in both the cytoplasm and the nucleus in the cIAP2[H570A] MEFs (Supplementary Fig. 12). The finding that RARγ localizes in both the cytoplasm and the nucleus in L929 cells (Supplementary Fig. 9) provides new insights about why TNF induces necroptosis in these cells without the presence of Smac mimetic. In addition, while RA promotes the nuclear localization of RARs, RA pre-treatment has little effect on cell death and the cytoplasmic localization of RARγ in response to TSZ in HT-29 cells, but however, RA pre-treatment did indeed partially block TNF-induced necroptosis in L929 cells (Supplementary Fig. 23).

Our study reveals a function for the nuclear receptor RARγ as a key regulator of RIP1-initiated cell death when cIAP activity is inhibited. As RIP1-initiated cell death is a vital cellular response triggered by death factors, the engagement of this pathway is finely regulated. As shown in Fig. 8b, the requirement of RARγ, which is released from the nucleus, provides a critical checkpoint for the transition from survival signaling to death machinery of RIP1-initiated cell death. This safety control by RARγ allows cells to engage death pathways only when they are fully committed to die.

## Methods

**Reagents and antibodies**. TNF-α and z-VAD-fmk were purchased from R&D. Cyclohexamide (CHX) from Sigma. Smac mimetic was a gift from S. Wang (University of Michigan, Ann Arbor, Michigan, USA). TRAIL from Novoprotein. FasL from Upstate Biotech. CH11 from MBL Medical and Biol Lab.

All antibodies are at concentration of 1 μg ml$^{-1}$ and used at 1:1,000 dilution unless otherwise stated. Anti-RARγ (C-15) (sc-550) for human, anti-RARα (C-20) (sc-551), anti-caspase-8 (C-20) (sc-6136), anti-cIAP2 (sc7944) and anti-Fas (C-20) (sc-715) from Santa Cruz; anti-RIP1 (610459) and anti-FADD (610400) from BD Biosciences; anti-RARγ1 (ab5905) for mouse, anti-RIP3 (ab72106), anti-MLKL (ab184718) for human, anti-MLKL (ab172868) for mouse and anti-cIAP1 (ab2399) from Abcam; anti-RIP3 (2283) for mouse from ProSci, anti-TRADD (05-473) from Upstate; anti-TRAF2 (MAB3277) and anti-TNFR1 (AF-425-PB) from R&D; anti-Actin (A3853) (dilution 1:10,000), anti-FLAG (F9291) (dilution 1:5,000) and anti-GFP (G6539) (dilution 1:5 000) from Sigma; anti-V5 (R960-25) (dilution 1:5,000) from Invitrogen; anti-PARP1 (BML-SA250-0050) from Enzo Life Science; anti-GAPDH (NB300-22) (dilution 1:5,000) from Novus Biologicals; anti-cleaved caspase-8 (9496), anti-CYLD (4495), anti-RIP1 (137451) and anti-p-RIP1(65746) from Cell Signaling Technology; anti-DsRed (632392) (dilution 1:5,000) from Clontech.

**shRNA library screening**. shRNA screening was performed using a retroviral shRNA library targeting cancer-related genes[33]. HT-29 cells were cultured in 15 cm dishes and infected using 8 μg ml$^{-1}$ polybrene (Millipore). Duplicated infections were carried out. At 24 h after the infection, media was replaced with 2 μg ml$^{-1}$ puromycin. After a 4-day puromycin selection, cells were treated with TSZ (TNF-α 30 ng ml$^{-1}$, Smac 10 nM and z-VAD-fmk 20 μM) for 48 h. The media were replaced with fresh completed medium containing 2 μg ml$^{-1}$ puromycin. Two days later, the cells were treated with TSZ for an additional 48 h. The colonies were picked and the genomic DNA was isolated for PCR and DNA sequencing analysis. shRNA (5′-ATAAATA GAGGCTTCCT CTGG-3′) targeting human rarγ gene was identified. The firefly luciferase shRNA (cont-shRNA) was used as a negative control shRNA.

**Generation of RARγ1 knockout mice by CRISPR**. All animal experimental protocols were performed in accordance to the NIH Animal Care & Use Committee guidelines. The RARγ knockout mice were generated using the CRISPR/Cas9[44]. Two sgRNAs were designed to target the 1st coding exon (Exon 3) of the mouse rarγ gene, one (5′-aaggagagactctttgcgccCGG-3′) targeted shortly after the translation initiation codon (ATG) and the other one (5′-gggccagcctgacctccc-caAGG-3′) targeted near the end of Exon 3. The two sgRNA DNA constructs were made using OriGene's gRNA Cloning Services (Rockville, Maryland) and were then used as templates to synthesize sgRNAs using MEGAshortscript T7 Kit (Invitrogen). The two sgRNAs were co-microinjected with Cas 9 mRNA into the cytoplasm of fertilized mouse eggs at the concentration of 100 μg ml$^{-1}$ of Cas9 and 20 μg ml$^{-1}$ of each sgRNA. The injected zygotes were cultured overnight in M16 medium. Those embryos reached 2-cell stage of development were implanted into the oviducts of pseudopregnant surrogate mothers. Mice born to these foster mothers were genotyped by PCR amplification of the region surrounding the CRISPR cutting site followed by DNA sequencing. Founder mice with X and Y base pair deletions were bred with wt C57BL/6 mice to establish two independent RARγ knockout mouse lines, which were used in subsequent studies.

**Cell culture and treatment**. HT-29, MCF-7, HeLa, HEK293T, L929 and MEFs cells were purchased from ATCC and maintained in Dulbecco's modified Eagle's medium (DMEM) containing 4.5 g ml$^{-1}$ glucose, 10 % fetal bovine serum and 2 mM L-glutamine with 100 units penicillin per ml and 100 units streptomycin per ml.

Cell death treatment: Necroptosis was induced by TNF-α (30 ng ml$^{-1}$), Smac (10 nM) and z-VAD-fmk (20 μM) or TNF-α (30 ng ml$^{-1}$), CHX (10 μg ml$^{-1}$) and z-VAD-fmk (20 μM). Apoptosis was induced by TNF-α (30 ng ml$^{-1}$) and Smac (10 nM) or TNF-α (30 ng ml$^{-1}$), CHX (10 μg ml$^{-1}$). Necroptosis was induced by TRAIL (250 ng ml$^{-1}$), FasL (2 ng ml$^{-1}$) or CH11 (100 ng ml$^{-1}$) with Smac (10 nM) and z-VAD-fmk (20 μM) treatment for 24 h, respectively. Apoptosis was induced by TRAIL (250 ng ml$^{-1}$), FasL (2 ng ml$^{-1}$) or CH11 (100 ng ml$^{-1}$) with Smac (10 nM) for 24 h, respectively.

**Preparation of primary MEFs**. Primary MEFs were prepared as described by Pobezinskaya et al.[47]. Briefly, embryos at day 13.5 of gestation were isolated from

pregnant mice. Each embryo was incubated for 30 min at 37 °C in a separate Eppendorf tube with Cellstripper solution (Cellgro). Cells were gently 'mashed' by being pipetting, then filtered through cell strainers and cultured in DMEM with 10% (vol/vol) fetal bovine serum (FBS).

**Macrophage differentiation in vitro.** Bone marrow-derived monocytes were isolated from mouse femurs and tibias and cultured for 2 h. The attached monocyte enriched cells were cultured in RPMI-1640 medium supplemented with 10% (vol/vol) FBS and 2 mM glutamine, with penicillin (100 U ml⁻¹) and streptomycin (100 µg ml⁻¹). For differentiation, cells were cultured for 6 days in the presence of recombinant mouse M-CSF (20 ng ml⁻¹).

**Propidium iodide and AnnexinV staining.** Cells were washed and resuspended in HEPES buffer containing propidium iodide and/or Annexin V-fluorescein isothiocyanate (BD biosciences). The stained cells were analyzed with flow cytometry.

**Plasmid construction and shRNA.** Human RARγ transcript variant 1 (NM_000966) was cloned into the mammalian expression vector pcDNA4 V5/HisC, pEGFP-C1 and pFLAG-C1. Human RARγ was inserted into the lentiviral vector pLESIP to generate stable cell lines for the rescue experiments. All of the plasmid constructs were confirmed by DNA sequencing. lentiviral vector expressing RFP-RXRα were obtained from GeneCopoeia (Rockville, MD, USA).

RARγ NLS point mutants and confirmed by DNA sequencing. To generate RARγ NLS point mutants were generated using site-direct mutagenesis. Four amino acids substitution from lysine (K) to alanine (A) of amino acid 166 to 169 within the NLS sequence (RNKKKK, amino acid 164 to 169) were mutated using site-direct mutagenesis and confirmed by DNA sequencing.

The shRNA lentiviral plasmids were purchased from Sigma. The shRNA against human RARγ (NM_000966) corresponds to the coding regions 1,065–1,085 (RARγ sh1) and 1,344–1,364 (RARγ sh2). The shRNA against mouse RARγ (NM_011244) corresponds to the 3′-untranslated region 2,030–2,050 (RARγ shRNA). The shRNA against human RARα (NM_000964) corresponds to the coding region 547-567 (RARα sh1) and 1531-1551 (RARα sh2). The shRNA RARγ (NM_000966, RARγ shRNA-A, RARγ shRNA-B and RARγ shRNA-C) clones targeting the 3′-untranslated region 2,680–2,700 and the shRNA RIP1 (NM_003804, RIP1 shRNA) clone targeting the coding region 1,567–1,588 were obtained from the screening. siRNAs (SMARTpool) against mouse RARγ and scrambled siRNA (SMARTpool) were purchased from Dharmacon (Lafayette, CO, USA).

All plasmids were transfected into cells using Lipofectime-Plus reagents as per manufacturer's recommendations (Invitrogen).

**Lentiviral infection.** HEK293T were co-transfected with pCMV-VSV-G and pCMV-dr8.2-dvpr and either non-targeting-shRNA or RARγ-shRNA or RARα-shRNA plasmids. After 24 h, supernatant was collected and this lentiviral preparation was used to infect cells. After 24 h of infection, cells were selected with puromycin for a further 48 h.

For cell rescue assay, HT-29 expressing RARγ shRNA obtained from screening were infected with pLESIP-rRARγ or pLESIP-rRARγ-NLSmut.

**Immunoblotting crosslinking and immunoprecipitation.** Cell were collected and lysed in M2 buffer (20 mM Tris at pH7, 0.5% NP40, 250 mM NaCl, 3 mM EDTA, 3 mM EGTA, 2 mM dithiothreitol, 0.5 mM phenylmethylsulfonyl fluoride (PMSF), 20 mM β-glycerol phosphate, 1 mM sodium vanadate and 1 µg ml⁻¹ leupeptin). Cell lysates were separated by 4-20% polyacrylamide gel electrophoresis and analyzed by immunoblotting. The proteins were visualized by enhanced chemiluminescence according to the manufacture's (Amersham) instructions.

For immunoprecipitation, the lysates were precipitated with antibodies (1 µg) and protein-G agarose bead by incubation at 4 °C overnight. The beads were washed four to six times with 1 ml M2 buffer and the bound proteins were removed by boiling in SDS buffer and resolved in 4-20% SDS-polyacrylamide gels for western blot analysis.

For endogenous RARγ immunoprecipitation, crosslinking of cellular proteins was performed before the cell lysis. Cell were washed three times with PBS solution and incubated with a crosslinking reagent dithiobis(succinimidyl propionate) (DSP) in PBS for 30 min at room temperature followed by incubation in 10 mM Tris-HCl pH 7.5 buffer for 15 min to stop the reaction. The cells were lysed in cytosolic buffer (10 mM HEPES pH 7.9, 1.5 mM MgCl₂, 10 mM KCl, 0.05% NP40, 0.5 mM PMSF, 20 mM β-glycerol phosphate, 1 mM sodium vanadate and 1 µg ml⁻¹ leupeptin) to remove nuclear proteins. The lysates were incubated with 2 µg of RARγ (C-15) antibody (sc-550, Santa Cruz) at 4 °C overnight followed by incubation with protein-G agarose beads at 4 °C for 4 h. Then, the samples were used in regular immunoprecipitation procedure.

The images of uncropped gels reported in the main figures and Supplementary Figures are shown in Supplementary Figs 24–29.

**Confocal imaging and analysis.** All confocal images were visualized using a Carl Zeiss LSM780 confocal microscope equipped with a Plan-Apochromat 63 × numerical aperture 1.40 DIC oil objective. Images were acquired and analyzed

using Carl Zeiss ZEN software. HeLa cells were cultured in 3-cm Ibidi plates (Ibidi) for 24 h and transfected with 1 µg pEGFP-RARγ by Lipofectamine-Plus reagent (Invitrogen). For overexpression proteins, 24 h post transfection, cells were treated with dimethyl sulfoxide (DMSO), TNF-α or TS for 2 h and stained with nuclear 4′,6-diamidino-2-phenylindole (DAPI) and directly visualized by confocal microscopy. For endogenous proteins, HT-29 cells were treated with DMSO or TSZ for 2 h. Cells were fixed and stained with anti-F-actin and anti- RARγ antibodies, and nuclear DAPI and visualized by confocal microscopy.

**Mouse model of systemic inflammation.** mTNF-α and inhibitors were diluted in endotoxin-free PBS and injected in a volume of 0.2 ml. mTNF-α were injected intravenously (i.v.), whereas z-VAD-fmk was injected intraperitoneally (i.p.). Z-VAD-fmk was given 15 min before (250 mg) and 1 h after (10 mg) mTNF. Liver were collected at designated times after injection. RARγ –/– and wt siblings 7–8 weeks of age were sensitized by intraperitoneal administration of 700 mg GalN per kg body weight (700 mg kg⁻¹). After 15 min of GalN treatment, a sublethal dose of recombinant mouse TNF diluted with pyrogen-free saline (20 mg/kg) was administered i.v.

**Statistical analysis.** Statistical analyses were performed using GraphPad Prism 5 Software. Comparisons between two groups were made by Student's t-test, and comparisons between more than two groups were made by one-way analysis of variance. Survival curves (Kaplan–Meier plots) were compared using a log rank test. All P-values less than 0.05 were considered statistically significant.

**Data availability.** The authors declare that all data supporting the findings of this study are available within the article and its Supplementary Information Files or from the corresponding author upon reasonable request.

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

## Acknowledgements

We thank Dr. Ji Luo for the cancer-related gene shRNA library and Dr. Jonathan Ashwell for cIAP1−/− and cIAP2$^{H570A}$ MEFs. We thank Ross Lake for his expertise in confocal imaging. This research was supported by the Intramural Research Program of the Center for Cancer Research, National Cancer Institute, National Institutes of Health.

## Author contributions

Q.X. and S.J. designed and performed most of the experiments. C.K. and S.C. conducted mouse experiments and helped to supervise the project. S.J., Q.X. and S.C. wrote parts of the manuscript. C.K. J.Q., J.J. and M.C. helped with experiments. C.L. generated RARγ1 knockout mice. Z.-g.L. conceived, supervised and directed the project and wrote the manuscript.

## Additional information

**Competing interests:** The authors declare no competing financial interests.

