## [Peer Review File · Nature Communications]

This manuscript has been previously reviewed at another journal that is not operating a transparent peer review scheme. This document only contains reviewer comments and rebuttal letters for versions considered at Nature Communications.

Reviewers' Comments:

Reviewer #1:

Remarks to the Author:

Xu et al have now addressed many of the points raised previously.

I support publication after minor revision:

- 1) The title should be modified because not all RIP1-dependent deaths required RARg. See Fig. S2b – RIP1 loss protected against TCZ, RARg loss did not.
- 2) Line 179 should be corrected to "RIP3 autophosphorylation is an early event in TNF-induced necroptosis." The Francis Chan Cell paper that is referenced examined whether RIP1 might phosphorylate RIP3, but found no evidence for this. To date, the only known RIP1 substrate is RIP1 itself.
- 3) A WB for MLKL might be included in fig. 1a.

Reviewer #2:

Remarks to the Author:

Xu et al. discovered the role of RARγ in mediating apoptosis induced by T/S and necroptosis induced by T/S/Z. The authors showed that RARγ was released to the cytoplasm from the nucleus during RIP1-initiated apoptosis and necroptosis. The authors conclude that RARγ regulate RIP1-dependent cell death.

Despite the efforts in revision by the authors, the mechanism remains confusing due to following reasons:

- 1) The authors propose that RARγ regulates RIP1-dependent cell death, but it has no effect on TCZ induced necroptosis, which is also RIP1-dependent.
- 2) The authors did not use now well-established p-abs for RIPK1, RIPK3 and MLKL to demonstrate the effect of RARγ knockdown on necroptosis. These abs are commercially available from CST and Abcam.
- 3) The authors should examine the effect of RARγ knockdown to TSZ using CellTiterGlo. %PI+ by FCAS can be an ambiguous assay: e.g. does 80% PI+ really mean that 80% of cells died?
- 4) The authors should examine the effect of RARγ on cIAP/2 levels. If RARγ reduced the cellular levels or localization of cIAP1/2 might also explain the resistance to TSZ.

Reviewer #3:

Remarks to the Author:

This manuscript has been significantly improved after the revisions, and is now suitable for publication with one minor revision:

- The authors should clearly indicate the non-genomic function of cytoplasmic RARγ in RIP1-initiated death in the Abstract.

Our point-to-point responses to the reviewers are the following:

Reviewer #1 (Remarks to the Author):

Xu et al have now addressed many of the points raised previously.

I support publication after minor revision:

- 1) The title should be modified because not all RIP1-dependent deaths required RAR γ . See Fig. S2b – RIP1 loss protected against TCZ, RAR γ loss did not.

Response: we have changed our title to tone down our conclusion.

- 2) Line 179 should be corrected to “RIP3 autophosphorylation is an early event in TNF-induced necroptosis.” The Francis Chan Cell paper that is referenced examined whether RIP1 might phosphorylate RIP3, but found no evidence for this. To date, the only known RIP1 substrate is RIP1 itself.

Response: we made the correction as the reviewer suggested in our text.

- 3) A WB for MLKL might be included in fig. 1a.

Response: we added a western blot for MLKL in Fig 1a.

Reviewer #2 (Remarks to the Author):

Xu et al. discovered the role of RAR γ in mediating apoptosis induced by T/S and necroptosis induced by T/S/Z. The authors showed that RAR γ was released to the cytoplasm from the nucleus during RIP1-initiated apoptosis and necroptosis. The authors conclude that RAR γ regulate RIP1-dependent cell death.

Despite the efforts in revision by the authors, the mechanism remains confusing due to following reasons:

- 1) The authors propose that RAR γ regulates RIP1-dependent cell death, but it has no effect on TCZ induced necroptosis, which is also RIP1-dependent.

Response: to tone down our conclusion by excluding TCZ-induced necroptosis, we changed our title.

- 2) The authors did not use now well-established p-abs for RIPK1, RIPK3 and MLKL to demonstrate the effect of RAR γ knockdown on necroptosis. These abs are commercially available from CST and Abcam.

Response: we already changed RIP-1 phosphorylation with the anti-p-RIP1 antibody Suppl. Fig 3B). The RIP3 phosphorylation was shown by its mobility change, which has been used by many

previous publications. Therefore, it is not necessary to use the anti-RIP3 antibody. Since we showed MLKL is not recruited to RIP1/3 complex, it is not necessary to check its phosphorylation by RIP3.

- 3) The authors should examine the effect of RAR γ knockdown to TSZ using CellTiterGlo. %PI+ by FCAS can be an ambiguous assay: e.g. does 80% PI+ really mean that 80% of cells died?

Response: PI staining is a widely used, reliable method to check cell death with cell membrane permeability. In contrast, the reviewer suggested the method, CelltiterGlo, is to ATP levels of dying cells. Since many factors could affect cell ATP levels, such as mitochondrial function loss, this method is not a specific one for measuring cell death.

- 4) The authors should examine the effect of RAR γ on cIAP/2 levels. If RAR γ reduced the cellular levels or localization of cIAP1/2 might also explain the resistance to TSZ.

We have shown that RAR γ works downstream of cIAPs inhibition. So, the reviewer's question is not valid since loss of IAP is a necessary event, upstream of RAR γ . Reducing cIAP levels only promotes cell death, not causes resistance.

Reviewer #3 (Remarks to the Author):

This manuscript has been significantly improved after the revisions, and is now suitable for publication with one minor revision:

- The authors should clearly indicate the non-genomic function of cytoplasmic RAR γ in RIP1-initiated death in the Abstract.

Response: we made the change in our abstract.